biochemistry/molecular biology/cellular biology

connexin, connexon (hemichannel), mRNA, alternative splicing, keratitis ichthyosis deafness syndrome

**Author for correspondence:**
Nicholas Dale
e-mail: n.e.dale@warwick.ac.uk

# Cx26 keratitis ichthyosis deafness syndrome mutations trigger alternative splicing of Cx26 to prevent expression and cause toxicity *in vitro*

## Jonathan Cook, Elizabeth de Wolf and Nicholas Dale

School of Life Sciences, University of Warwick, Coventry CV4 7AL, UK

ND, 0000-0003-2196-2949

The Cx26 mRNA has not been reported to undergo alternative splicing. In expressing a series of human keratitis ichthyosis deafness (KID) syndrome mutations of Cx26 (A88V, N14K and A40V), we found the production of a truncated mRNA product. These mutations, although not creating a cryptic splice site, appeared to activate a pre-existing cryptic splice site. The alternative splicing of the mutant Cx26 mRNA could be prevented by mutating the predicted 3′, 5′ splice sites and the branch point. The presence of a C-terminal fluorescent protein tag (mCherry or Clover) was necessary for this alternative splicing to occur. Strangely, $Cx26^{A88V}$ could cause the alternative splicing of co-expressed WT Cx26—suggesting a trans effect. The alternative splicing of $Cx26^{A88V}$ caused cell death, and this could be prevented by the 3′, 5′ and branch point mutations. Expression of the KID syndrome mutants could be rescued by combining them with removal of the 5′ splice site. We used this strategy to enable expression of $Cx26^{A40V-5′}$ and demonstrate that this KID syndrome mutation removed $CO_2$ sensitivity from the Cx26 hemichannel. This is the fourth KID syndrome mutation found to abolish the $CO_2$-sensitivity of the Cx26 hemichannel, and suggests that the altered $CO_2$-sensitivity could contribute to the pathology of this mutation. Future research on KID syndrome mutations should take care to avoid using a C-terminal tag to track cellular localization and expression or if this is unavoidable, combine this mutation with removal of the 5′ splice site.

# 1. Introduction

The human connexin26 (Cx26) gene is comprised of a short non-coding exon 1, followed by a long intron and a long protein-coding exon 2. The promotor P1, located upstream of exon 1, contains a TATA motif (TTAAAA) and several GC boxes for specificity protein 1 (Sp1) and specificity protein 3 (Sp3) binding [1–3]. Connexin gene expression is predominantly controlled by transcriptional regulation, in a tissue-dependent manner [4]. The resulting Cx26 mRNA contains an internal ribosome entry site (IRES) in the 5′ untranslated region and a signal for the initiation of cap-independent translation [4,5]. Alternative splicing of Cx26 has never been reported.

Like all connexins, six protein subunits assemble to form a hexameric Cx26 hemichannel or connexon. Hemichannels in opposing cell membranes dock together to form a gap junction, facilitating the transmission of small molecules, including ions, metabolites and signalling molecules. Cx26 also exhibits functionality independently of gap junctions; hemichannels are opened in response to an increase in $PCO_2$, resulting in the release of ATP, which acts on respiratory neurons to stimulate breathing [6,7]. We demonstrated, by mutational analysis, that $CO_2$ directly led to the carbamylation of residue K125, resulting in the formation of a salt bridge to R104 in an adjacent subunit of the connexin hexamer [8].

Mutations in the Cx26 gene have been associated with human pathology of varying severity, ranging from non-syndromic deafness to keratitis ichthyosis deafness (KID) syndrome, a disease characterized by skin lesions, sensorineural hearing loss and vascularizing keratitis [9–18]. Nine mutations in Cx26 have been linked to KID syndrome [19]. One hypothesis to account for the pathological nature of these dominant mutations is that they give rise to leaky hemichannels which, being less sensitive to $Ca^{2+}$ block, are open under resting conditions thereby causing cell death [20–24]. To some extent altered Cx26 gating properties of some mutations may correlate with the severity of the pathology [25]. The A88V mutation in Cx26, which causes a very severe form of KID syndrome, can also occur in the related connexin Cx30 where it causes Clouston's syndrome [26,27]. Recently Cx30[A88V], although being toxic to cells in culture, was found to confer protection against age-related hearing loss *in vivo* [28] suggesting caution in extrapolation of effects from *in vitro* model systems to whole animal physiology.

We have recently reported that Cx26 hemichannels with N14K or N14Y mutations do not open in response to $CO_2$ challenge [29]. We previously reported that the A88V mutation in rat Cx26 removed $CO_2$ sensitivity in a dominant-negative fashion [30]. Both N14Y and N14K exert powerful dominant-negative action on the $CO_2$ sensitivity of Cx26[WT] [29]. In this paper, we have explored other KID syndrome mutations (A40V and A88V in human Cx26) that we had been previously unable to express in HeLa cells. Surprisingly, we found the existence of pre-existing splicing at cryptic splice sites within Cx26 that became active when Cx26 contained certain KID syndrome mutations. Once splicing had been prevented, expression of the Cx26 KID syndrome mutant proteins was restored. We used this method to show that the KID syndrome mutation A40 V (in human), like A88V (in rat), N14K and N14Y (in rat), abolished the sensitivity of Cx26 to $CO_2$. This cryptic splicing seemed to depend on the existence of a C-terminal fluorescent protein tag, suggesting that this method of localizing Cx26 expression is not suitable for KID syndrome mutants.

# 2. Material and methods

## 2.1. Connexin 26 mutagenesis

The sequence for wild-type Cx26 was taken from accession number: NM_004004.5. Mutations were introduced into the human Cx26 sequence [29] by the Quikchange protocol (Agilent) using the following primers:

A88V: for 5′ cgtgtccacgccagTgctcctagtggcc 3′, rev 5′ ggccactaggagcactggcgtggacacg 3′;
A40V: for 5′ ctcgttgtggctgTaaaggaggtgtgg 3′, rev 5′ ccacacctcctttacagccacaacgag 3′;
N14K: for 5′ ctggggggtgtgtgaaGaaacactccaccag 3′, rev 5′ ctggtggagtgtttcttcacaccccccag 3′;
A88S: for 5′ cgtgtccacgccaTcgctcctagtggcc 3′, rev 5′ ggccactaggagcgatggcgtggacacg 3′.

Cx26 was then cloned, in frame, into the pCAG-GS mCherry vector. Similar constructs were also made with a pcDNA 3.1 mCherry vector. Splice site mutations were made with the following primers:

3′ splice site mutant 646C: for 5′ gttatttgctaattCgatattgttctgggaag 3′, rev 5′ cttcccagaacaatatcgaattagcaaa
taac 3′, [646]AGA (arginine) silently mutated to [646]CGA (arginine);

Branch point mutants: for 5′ gtgtctggaatttgcatcctgctgCaGgtcactCTattgtgttatttgctaattagatattg 3′, rev 5′ caatatctaattagcaaataacacaatagagtgacctgcagcaggatgcaaattccagacac 3′, [616]AAT (asparagine) mutated to [616]CAG (glutamine) and [625]GAA (glutamic acid) mutated to [625]CTA (leucine);

5′ splice site mutant M151 L: for 5′ catcttcgaagccgccttcCtCtacgtcttctatgtcatgt 3′, rev 5′ acatgacatagaagacgt agaggaaggcggcttcgaagatg 3′, [451]ATG (methionine) mutated to [451]CTC (leucine).

The presence of the correct Cx26 sequence in each construct was confirmed by DNA sequencing (GATC Biotech).

## 2.2. Hela cell culture and transfection

HeLa Ohio (ECACC) cells were grown in EMEM supplemented with 1% non-essential amino acids, 10% fetal bovine serum, 50 µg ml$^{-1}$ penicillin/streptomycin and 3 mM CaCl$_2$. For microscopy and dye loading experiments, cells were plated onto coverslips at a density of $1 \times 10^5$ cells per well of a six-well plate. The cells were transiently transfected with Cx26 constructs containing a fluorescent marker, mCherry (1 µg) as described in the GeneJuice Transfection Reagent protocol (Merck Millipore). Cx26 expression was determined by imaging mCherry fluorescence using an Axiovert A1 microscope.

## 2.3. cDNA preparation

HeLa cells were harvested 48 h post-transfection and washed twice with cold PBS. The cell pellet was stored at −80°C. Total RNA was prepared using the Reliaprep RNA Cell Miniprep System (Promega). cDNA was generated with GoScript reagents (Promega) using random hexamer primers. Taq polymerase (Invitrogen) was used to PCR Cx26 from the cDNA. The forward primer annealed upstream of the start codon (ngF for pCAG-GS and pcfor for pcDNA 3.1) and the reverse primer annealed at the 3′ end of the Cx26 coding sequence (tag-cxRev).

pCAG-GS: ngF 5′ GCAACGTGCTGGTTATTGTGCTGT 3′;
pcDNA 3.1: pcfor 5′ CACTGCTTACTGGCTTATCG 3′;
Tag-cxRev: 5′ tcatggtggcgaataaCTGGCTTTTTTGACTTCC 3′.

## 2.4. Dye loading experiments

Our dye loading protocol has been described previously [7,8,29–32]. HeLa Ohio cells expressing either Cx26$^{WT}$, Cx26$^{WT-5'}$ or Cx26$^{A40V-5'}$ for 48 h were exposed to control, zero Ca$^{2+}$ or hypercapnic aCSF containing 200 µM 5(6)-carboxyfluorescein (CBF) for 10 min. Cells were then exposed to control aCSF with 200 µM CBF for a further 5 min, followed by a wash in control aCSF with no CBF for 30 min to remove excess extracellular dye. A replacement coverslip of HeLa cells was used for each condition.

## 2.5. Artificial cerebrospinal fluid solutions

### 2.5.1. Control aCSF

124 mM NaCl, 26 mM NaHCO$_3$, 1.25 mM NaH$_2$PO$_4$, 3 mM KCl, 10 mM D-glucose, 1 mM MgSO$_4$, 1 mM CaCl$_2$. This was bubbled with 95%O$_2$/5% CO$_2$ and had a final pH of approximately 7.4.

### 2.5.2. Zero Ca$^{2+}$ aCSF

124 mM NaCl, 26 mM NaHCO$_3$, 1.25 mM NaH$_2$PO$_4$, 3 mM KCl, 10 mM D-glucose, 1 mM MgSO$_4$, 1 mM MgCl$_2$, 1 mM EGTA. This was bubbled with 95%O$_2$/5% CO$_2$ and had a final pH of approximately 7.4.

### 2.5.3. Hypercapnic (55 mmHg CO$_2$) aCSF

100 mM NaCl, 50 mM NaHCO$_3$, 1.25 mM NaH$_2$PO$_4$, 3 mM KCl, 10 mM D-glucose, 1 mM MgSO$_4$, 1 mM CaCl$_2$. This was bubbled with sufficient CO$_2$ (approx. 9%, balance O$_2$) to give a final pH of approximately 7.4.

## 2.6. Assessment of cell death

Cells plated at a density of $1 \times 10^5$ cells per well in a six-well plate were transfected as above. After 48 h, the media was collected, and a PBS wash (superfusate) also collected in the same tube. Cells were then harvested with trypsin and added to the tube containing the media and PBS superfusate. The cells were then pelleted, washed in fresh media, and the final pellet was resuspended in 2 ml of media. An aliquot was mixed 1 : 1 with trypan blue. Using a haemocytometer, the cells were counted to determine the proportion of dead (blue) cells. This was expressed relative to the proportion of dead cells found in similar cultures of dummy-transfected parental HeLa cells, that were treated with GeneJuice but no DNA.

## 2.7. Fluorescence imaging and analysis

After dye loading, HeLa cells were imaged by epifluorescence (Scientifica Slice Scope, Cairn Research OptoLED illumination, 60× water Olympus immersion objective, NA 1.0, Hamamatsu ImagEM EM-CCD camera, Metafluor software). The extent of dye loading was determined by drawing a region of interest (ROI) around each cell in ImageJ (Wayne Rasband, National Institutes of Health, USA) [33], and subsequently, measuring the mean pixel intensity of the ROI. The mean pixel intensity of a representative background ROI for each image was subtracted from each cell measurement from the same image. At least 50 cells expressing each Cx26-mCherry construct were analysed for each condition per experiment, and five independent repetitions using the same batch of cells were completed. Data were plotted both as cumulative probabilities (showing every data point) and box and whisker plots where the box is interquartile rage, bar is median, and whisker extends to the most extreme data point that is no more than 1.5 times the interquartile range. Points that fall outside this range are plotted individually as circles. To avoid pseudoreplication, the medians of the individual replicates were compared between conditions using the Mann–Whitney $U$ test with one-tailed comparisons.

# 3. Results

## 3.1. Keratitis ichthyosis deafness syndrome mutations cause the expression of two transcripts from the Cx26 gene

We introduced the KID syndrome mutations, A88V, A40V and N14K, along with A88S, a Cx26 mutant that has no syndromic pathology but may cause deafness [34,35], into human Cx26 linked to the mCherry fluorophore. We then expressed these mutant hemichannels in HeLa cells. After 24–72 h, punctate expression of Cx26$^{WT}$ and Cx26$^{A88S}$ was observed, however, no expression was present in viable cells transfected with Cx26 carrying the A88V (figure 1a), A40V or N14K mutations (data not shown). To determine if the Cx26 mRNA was present, total RNA was harvested from HeLa cells 48 h after transfection with Cx26 and cDNA was synthesized using random hexamer primers. PCR of Cx26 from the cDNA revealed two PCR products of approximately 550 bp and 660 bp size from HeLa cells transfected with A88V, A40V or N14K Cx26 (figure 1b). Only the larger PCR product (660 bp), corresponding to the full Cx26 product, was present in cells expressing WT or A88S Cx26 (figure 1b). We also noted that many cells expressing the KID syndrome mutated Cx26 appeared to undergo cell death.

The smaller PCR product (550 bp, Cx26Δ) from cells transfected with Cx26$^{A88V}$ was isolated and sequenced. Alignment of this Cx26Δ with Cx26 revealed a deletion from bases 453 to 647 in Cx26Δ (figure 2). This truncated molecule lacks transmembrane domain 4, the second extracellular loop and part of transmembrane domain 3. We reasoned that alternative splicing of the Cx26 mRNA may have resulted in the smaller product, and employed a human splicing finder [36] to identify potential cryptic splice sites. Such sites were predicted at the 5′ and 3′ end of the deletion (figure 2). Two possible branch points were predicted to be located in close proximity to the 3′ splice site (bases 611–618 and 621–627). Note that the KID syndrome mutations did not introduce splice sites, but instead seem to have enabled pre-existing cryptic splice sites to become active.

## 3.2. Analysis of the alternative splicing of the Cx26 mRNA

To determine if the splicing of the Cx26 mRNA could be prevented, each of the required features (5′ donor site, 3′ acceptor site and branch point) were mutated in turn. Firstly, the arginine

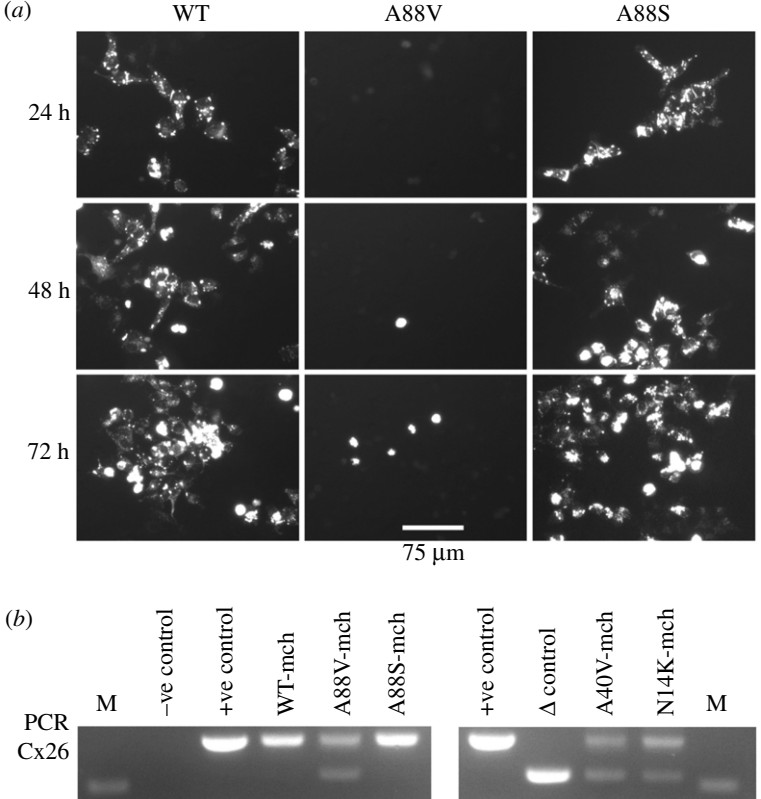

**Figure 1.** Reduced expression of Cx26$^{A88V}$ compared to Cx26$^{WT}$ and Cx26$^{A88S}$ in HeLa cells. (*a*) HeLa cells transfected with pCAG Cx26-mCherry. Representative images of mCherry fluorescence for each time point post-transfection. Strong punctate expression is seen for the WT and the A88S mutant, but not for the A88V mutant. (*b*) Total RNA was harvested at 48 h and cDNA made with random primers followed by PCR for Cx26. Negative control, cells transfected with no DNA; +ve control, Cx26-mCherry plasmid DNA; Δ control, Cx26Δ(453-647)-mCherry plasmid DNA. DNA size marker (M) is 500 bp.

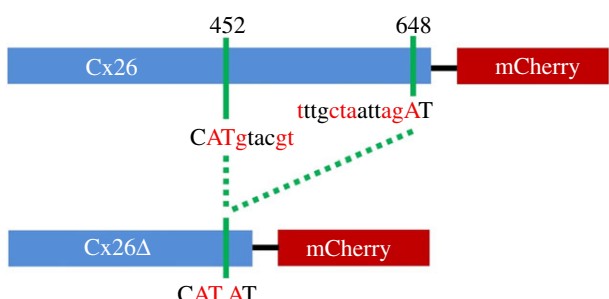

**Figure 2.** Cryptic splicing of Cx26-mCherry. Diagram of the cryptic splicing that results in the deletion product (not to scale). Indicated are the identified 5′ and 3′ sequences with lower case sequences in the cryptic intron and red and black denoting consecutive codons. The ATG codon (451-453) encodes M151.

codon AGA across the 3′ site was silently mutated to CGA (see methods). This mutation prevented splicing, but did not result in expression of either A88V, A40V or N14K mutant Cx26 (figure 3*a*).

Silent changes could not be introduced into the two potential branch points in order to disrupt them. Mutating the first branch point resulted in the conservative amino acid change of N206Q. The second branch point could also not be mutated conservatively, so the change E209L (see methods) was introduced in order to maintain hydrophobicity as it was located in the 4th transmembrane domain of Cx26. Again, the cryptic splicing was prevented and this time expression of Cx26$^{A88V}$ was seen (figure 3*b*). Unfortunately, these mutations appeared to decrease the level of WT Cx26-mCherry (compare to figure 3*d*).

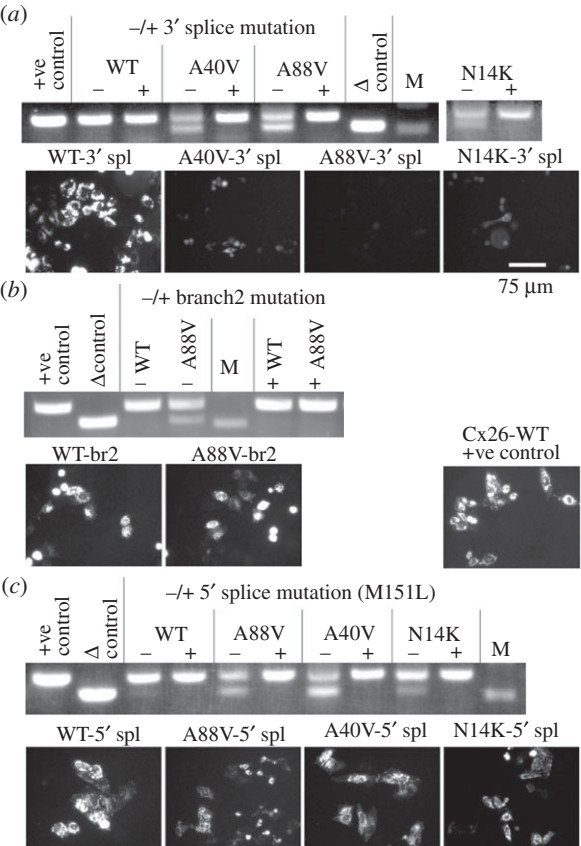

**Figure 3.** The effects on splicing and expression of splice site mutants in KID syndrome mutant Cx26. HeLa cells were transfected with variants of Cx26-mCherry DNA. Total RNA was harvested at 48 h. cDNA was made with random primers followed by PCR for cx26. +ve control is Cx26-mCherry plasmid DNA, Δ control is Cx26Δ(453-647)-mCherry plasmid DNA. DNA size marker (M) is 500 bp. (*a*) Mutation to remove cryptic 3′ splice site prevents splicing in all three KID syndrome mutations, but does not rescue expression of Cx26. (*b*) Mutations to remove both branch points in the A88V mutation prevent splicing and improve expression. (*c*) Mutation that removes the cryptic 5′ splice site prevents splicing and improves expression for all three KID syndrome mutations. Representative images shown of mCherry fluorescence at 48 h indicating expression levels of the splice site mutants. (*d*) Representative image of expression of Cx26WT +ve control for comparison to mutated Cx26 variants.

The 5′ site could not be mutated silently either, so a conservative M151L mutation (see methods) was introduced (located in the third transmembrane domain). This mutation abolished splicing and also allowed expression of A88V, A40V and N14K mutant Cx26 without appearing to affect the level of WT-Cx26 expression (figure 3*c* compared to 3*d*).

The simplest interpretation of our results is that the KID syndrome mutations somehow induce alternative splicing of the Cx26 mRNA at a cryptic splice site. This alternative splicing prevents expression of Cx26 and appears to have a toxic effect on the cells (WT versus A88V, figure 4). The 3′ splice mutations, although they prevented the splicing, did not rescue expression of the KID syndrome mutations.

We next examined what aspect of the alternative splicing caused cell toxicity by analysing the proportion of dead cells after transfection with WT Cx26 and Cx26$^{A88V}$ combined with the various mutations to stop splicing. Unlike expression of Cx26$^{WT}$ and its various splice site mutations or Cx26$^{A88S}$, expression of Cx26$^{A88V}$ caused a large increase in the proportion of dead cells (figure 4). A similar increase in cell death has been observed for the A88V mutation in Cx30 [28]. Cx30$^{A88V}$ also has reduced mRNA expression [28]. Whether this mutation triggers alternative splicing of the Cx30 mRNA in a manner analogous to what we report has not been investigated. All of the mutations that prevented splicing of Cx26$^{A88V}$ returned the proportion of dead cells to levels similar to Cx26$^{WT}$ (figure 4). As expression of the Cx26Δ did not induce significant cell death, the toxic effect must presumably be related to the mRNA itself and its ability to form the looped structure within the cells.

Curiously, the alternative splicing is only seen when the mutant channels are linked to the mCherry tag (figure 6*c,d*). Non-tagged Cx26$^{A88V}$ did not cause the elevated cell death seen with the mCherry-tagged construct (figure 4). We examined whether the ability to trigger alternative splicing

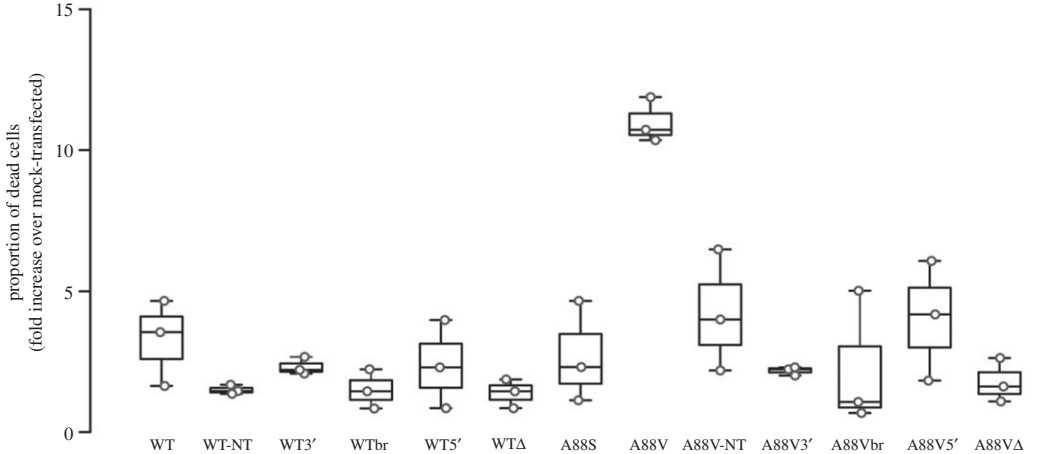

**Figure 4.** Cx26$^{A88V}$ induces cell death through alternative splicing. The proportion of dead cells revealed by a trypan blue assay relative to dummy-transfected parental cells (treated with GeneJuice but no DNA). Cx26$^{WT}$, its splice site mutations, truncated product Cx26$^{WT\Delta}$ and Cx26$^{A88S}$ exhibited the same low proportion of cell death. Cells expressing Cx26$^{A88V}$ exhibited a threefold increase in cell death compared to the cells expressing the Cx26$^{WT}$ and its variants. Prevention of the alternative splicing reduced the death-inducing effect of Cx26$^{A88V}$ to control levels. The truncated Cx26$^{A88V\Delta}$ product had no toxic effect. Note that untagged Cx26$^{A88V}$ (A88V-NT), which cannot be spliced, causes much less cell death than the tagged Cx26$^{A88V}$. 3′-mutated to remove 3′ splice site; 5′-mutated to remove 5′ splice site; br, both potential branch points mutated; NT, untagged construct. Box and whisker plot: the box is the interquartile range; the central line is the median; the whiskers are the range. Individual points are superimposed on the plot. Three independent replications for each mutant. All constructs tagged with mCherry. Three independent replicates for each variant.

was specific to the mCherry tag by testing Cx26$^{A88V}$ tagged with Clover (figure 5). We found the same phenomenon occurred—production of an alternative transcript that could be prevented by the 5′ splice mutation. Thus, the effect on splicing requires a fluorescent protein tag but is not specific to a particular fluorescent protein.

## 3.3. The keratitis ichthyosis deafness syndrome mutants cause alternative splicing of WT Cx26 in trans

The KID syndrome mutations are dominant. If this cryptic splicing were to contribute to the dominant pathogenic actions of the mutant Cx26 in KID syndrome one would predict that the presence of a KID syndrome mutant Cx26 might induce alternative splicing of WT Cx26 in trans (figure 6a), thus preventing expression of functional Cx26 and amplifying the proposed toxic moiety.

We, therefore, transfected HeLa cells with two different constructs. A 'reporter construct' that consisted of the WT Cx26 tagged with mCherry in the pCAG vector. The 'input construct' consisted of Cx26 (WT, variants of A88V, or A88S) tagged with mCherry in the pcDNA vector (figure 6a). The use of separate vectors allowed us to use vector-specific PCR primers to identify which mRNAs were being spliced—those of the input construct and/or those of the reporter construct. This strategy was verified by showing that the pCAG primers gave PCR products only with the pCAG construct and the pcDNA primers only with the pcDNA construct (figure 6b). When the input pcDNA construct had either WT Cx26 or Cx26$^{A88S}$, no alternative splicing was induced in the reporter construct (figure 6b). However, when the Cx26$^{A88V}$ was present, alternative splicing was induced in the reporter construct (figure 6b). Surprisingly, when the A88V mutation in the input construct was combined with further mutations of the 3′ or 5′ splice sites or the branch point, and thus underwent no splicing itself (figure 6b), splicing of the WT Cx26 was still induced in the reporter construct (figure 6b). This result shows that the splicing of the mutant Cx26 mRNA does not need to occur for the trans effect on splicing of the WT mRNA. It further implies that it is the conformation of the mutant input mRNA that mediates the effects in trans on the WT Cx26 mRNA of the reporter construct.

We have previously shown that the presence of an mCherry or Clover tag is required for this alternative splicing to manifest itself. We, therefore, tested whether a non-tagged input Cx26 variant might also induce alternative splicing in the WT Cx26 of the reporter (tagged with mCherry). Without the C-terminal mCherry tag, none of the mRNAs produced by the input construct were alternatively spliced

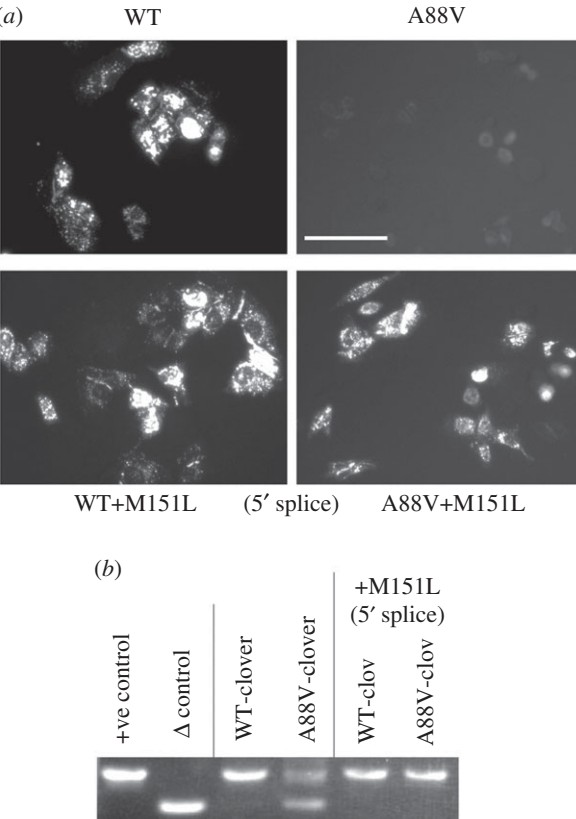

**Figure 5.** Cryptic splicing is also seen with clover-tagged Cx26. HeLa cells were transfected with variants of Cx26-clover DNA. (a) Representative images shown of clover fluorescence. White bar indicates 75 μm. (b) Total RNA was harvested at 48 h. cDNA was made with random primers followed by PCR for Cx26. +ve control is Cx26-clover plasmid DNA, Δ control is Cx26Δ(453-647)-mCherry plasmid DNA.

(figure 6c). However, all of the A88V variants (and to a very low extent A88S) induced alternative splicing in the pCAG reporter construct. Thus, the presence of the C-terminal tag is not required in the input mRNA to induce splicing. However, at least one of the Cx26 genes must be tagged—transfection of untagged Cx26$^{A88V}$ does not induce splicing in cells expressing untagged Cx26$^{WT}$ (figure 6d).

We then examined whether this trans-splicing action also affected the expression of WT Cx26 in HeLa cells. Like A88V, A88V-3′spl caused poor expression of the WT Cx26 (figure 7). This is perhaps not surprising as both these variants cause a high degree of alternative splicing of the WT Cx26 mRNA. More significantly A88V-5′spl allowed better expression of the WT Cx26 as did A88S (cf. figure 3). Although in these experiments both of these input transcripts caused a small amount of alternative splicing it was evidently not enough to prevent good expression of the WT Cx26.

## 3.4. The A40V mutation removes $CO_2$ sensitivity from Cx26 hemichannels

As we have previously shown that the A88V, N14K and N14Y mutations alter the $CO_2$ sensitivity of Cx26 in a dominant fashion [29,30], we determined the sensitivity of Cx26$^{A40V-5′}$ hemichannels in response to $CO_2$ via our previously established and validated dye-loading assay [7,8,29–32]. Removal of extracellular $Ca^{2+}$ results in opening of the hemichannels and, therefore, serves as a positive control as it reflects maximal dye loading into the cell [8,29,30]. HeLa cells expressing Cx26$^{WT}$ loaded with dye in response to increased $PCO_2$ (55 mmHg) and zero $Ca^{2+}$ conditions (figure 8a,b) as expected. Cells expressing Cx26$^{WT}$ with the 5′ splice site mutation (Cx26$^{WT-5′}$) also loaded with dye at $PCO_2$ of 55 mmHg and under zero $Ca^{2+}$ conditions (figure 8a,b). Furthermore, Cx26$^{A40V-5′}$ hemichannels also resulted in dye loading under zero $Ca^{2+}$ conditions, confirming that the functionality of the hemichannels was retained in the 5′ splice site mutants. HeLa cells expressing Cx26$^{A40V-5′}$ did not dye load at $PCO_2$ of 55 mmHg. As for the other KID syndrome mutations that we have tested [29,30], this mutation also abolishes the sensitivity of Cx26 to $CO_2$.

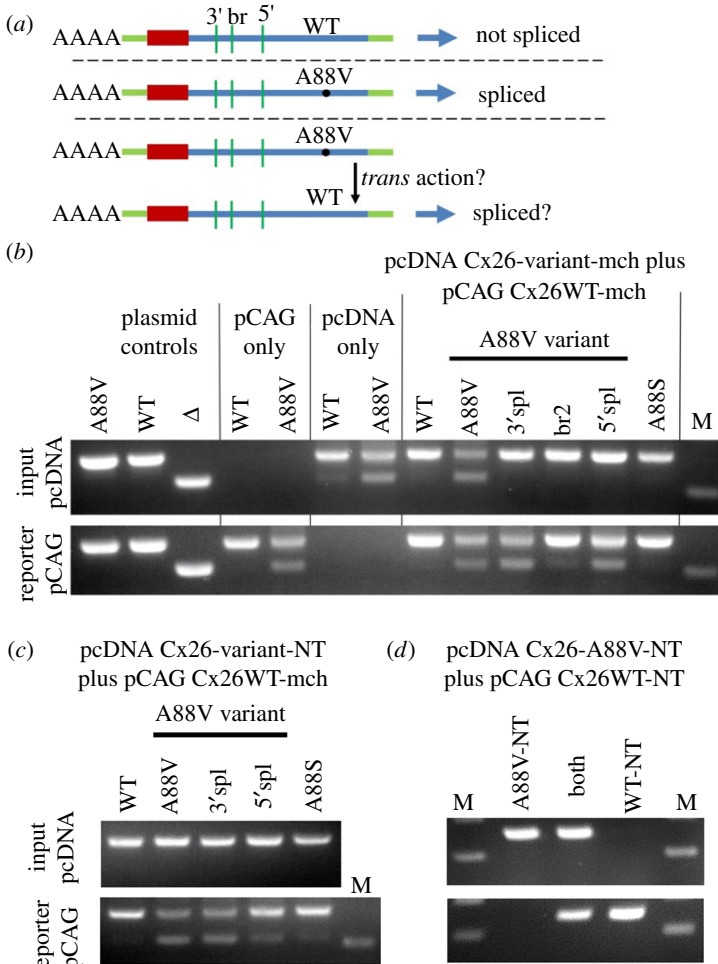

**Figure 6.** Cx26[A88V]-mCherry can cause Cx26[WT]-mCherry to splice in trans. (*a*) Diagram of the experimental logic. Light green, plasmid RNA; blue, Cx26 coding exon; red mCherry. (*b*) HeLa cells were cotransfected with the mCherry-tagged A88V variants, pcDNA Cx26-variant-mch (Input), and WT mCherry-tagged Cx26, pCAG Cx26WT-mch (Reporter). pCAG and pcDNA primer sets can only amplify sequences in their specific vectors. Cotransfection of the pcDNA-Cx26[A88V]-mCherry causes splicing of the pCAG-Cx26[WT] reporter, but the pcDNA-Cx26[A88S] does not cause this. The Cx26[A88V] does not need to splice itself to cause splicing in *trans* of Cx26[WT] -the 3′, 5′ and branch mutations prevent splicing of pcDNA-Cx26[A88V] but do not prevent splicing of the pCAG-Cx26[WT] reporter. (*c*) HeLa cells were cotransfected with the A88V variants lacking the mCherry tag (denoted by -NT), pcDNA Cx26-variant-NT (Input), and mCherry-tagged WT Cx26, pCAG Cx26WT-mch (Reporter). To induce splicing in *trans,* the input pcDNA-Cx26[A88V] does not need to carry the mCherry tag as long as the reporter pCAG-Cx26[WT] is tagged with mCherry. In the absence of the tag, the input pcDNA Cx26 variants show no sign of alternative splicing, however, the pCAG reporter Cx26 is spliced. Mutations to prevent splicing in the input pcDNA-Cx26[A88V] have no effect on the reporter splicing. (*d*) HeLa cells were cotransfected with the A88V lacking the mCherry tag, pcDNA Cx26-A88V-NT (Input), and untagged Cx26[WT], pCAG Cx26WT-NT (Reporter). No splicing is seen in either reporter or input mRNA as neither carries the mCherry tag. For all experiments, total RNA was harvested at 48 h. cDNA was made with random primers followed by PCR for Cx26 with a forward primer specific for the pCAG or pcDNA vectors and a reverse primer to the 3′ end of Cx26. Controls are Cx26-mCherry plasmid DNA and Δ control is Cx26Δ(453-647)-mCherry plasmid DNA. DNA size marker (M) is 500 bp.

# 4. Discussion

## 4.1. Implications of our results for *in vivo* and *in vitro* studies of keratitis ichthyosis deafness syndrome mutations

Our study has revealed the surprising observation that three KID syndrome mutations (A88V, N14K, A40V) induce a high degree of alternative splicing of the Cx26 mRNA. Interestingly, the non-syndromic mutation A88S does not induce this alternative splicing. The KID syndrome mutations

pCAG Cx26WT-mch cotransfected with:

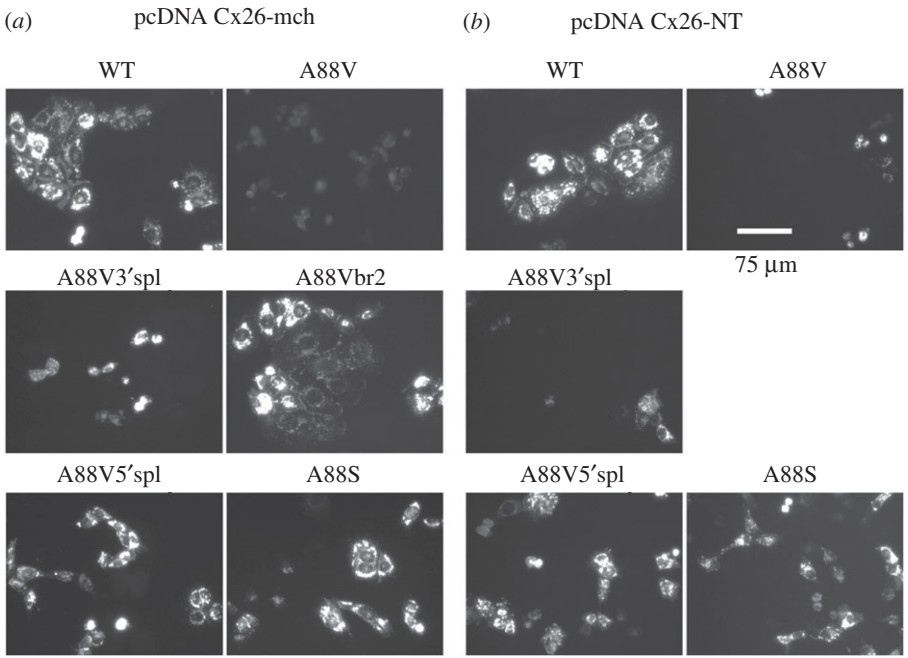

*(a)* pcDNA Cx26-mch    *(b)* pcDNA Cx26-NT

WT    A88V    WT    A88V

75 µm

A88V3′spl    A88Vbr2    A88V3′spl

A88V5′spl    A88S    A88V5′spl    A88S

**Figure 7.** The effect of Cx26^A88V on expression of Cx26^WT in *trans*. (*a*) HeLa cells were transfected with pCAG Cx26^WT-mCherry and several variants of pcDNA Cx26^A88V-mCherry. The mCherry fluorescence comes from both constructs. Strikingly little expression of either construct is seen when Cx26^WT and Cx26^A88V are cotransfected. (*b*) HeLa cells were transfected with pCAG Cx26^WT-mCherry and several variants of untagged pcDNA Cx26^A88V. In this case, the mCherry fluorescence can only arise from the pCAG Cx26^WT-mCherry. Expression of Cx26^WT is only seen with pcDNA Cx26^A88V-5′ and pcDNA Cx26^A88S.

do not create new splice sites, instead, they enable the pre-existing sites to become highly active. Occasionally, but not consistently, we observed a very low level of alternative splicing of the WT Cx26 mRNA on its own (e.g. figures 3*c* and 6*b*). As the cryptic splice site is present in the WT gene, this occasional very low level of alternative splicing is perhaps not unexpected, and clearly does not occur at a level that will affect the expression of the WT gene.

There are some important caveats to discuss before concluding this alternative splicing could mediate the pathological effects of these mutations *in vivo*. Firstly, the presence of a C-terminal fluorescent protein tag was required for the alternative splicing to occur. As the N-terminus of connexins is a short alpha-helix that is located within the pore of the channel [37], the alternative strategy of tagging connexins at the N-terminus is not an option. Secondly, our constructs lacked the non-translated regions that would be present in the real Cx26 mRNA transcribed *in vivo*. These non-translated regions could alter the stability and folding of the mRNA and potentially protect the mRNA from this aberrant splicing. Both of these caveats suggest that the splicing we report here might not occur *in vivo*. Nevertheless, this is speculation and we think it important and interesting to verify directly whether truncated Cx26 transcripts occur in mRNA extracted from KID syndrome patients.

Many investigators studying the properties of mutant hemichannels use a C-terminal tag to assess the expression and cellular localization of the hemichannel. There have been many reports of expression of KID syndrome mutations causing cell death. Our data suggest that if this is done in conjunction with a C-terminal fluorescent protein tag, the cell death may be connected to the alternative splicing of the mRNA rather than any property of the mutant hemichannel. At the very least, our data indicate that functional studies on KID syndrome Cx26 mutants should not be performed with a C-terminal tagged variant to avoid the effects of alternative splicing. However, if a C-terminal tag is desired, e.g. to study cellular localization or trafficking, then it should be combined with mutation of the 5′ splice site to prevent alternative splicing. While the mutation M151L that removes the 5′ splice site is not silent, M151 is in the 3rd transmembrane domain and is outward-facing [37]. The conservative substitution of a hydrophobic residue to interact with the membrane lipid is unlikely to cause significant conformational change. This is supported by our data showing that Cx26^M151L retains $CO_2$-sensitivity, permeability to the fluorescent dye (CBF) and can still be blocked by extracellular $Ca^{2+}$.

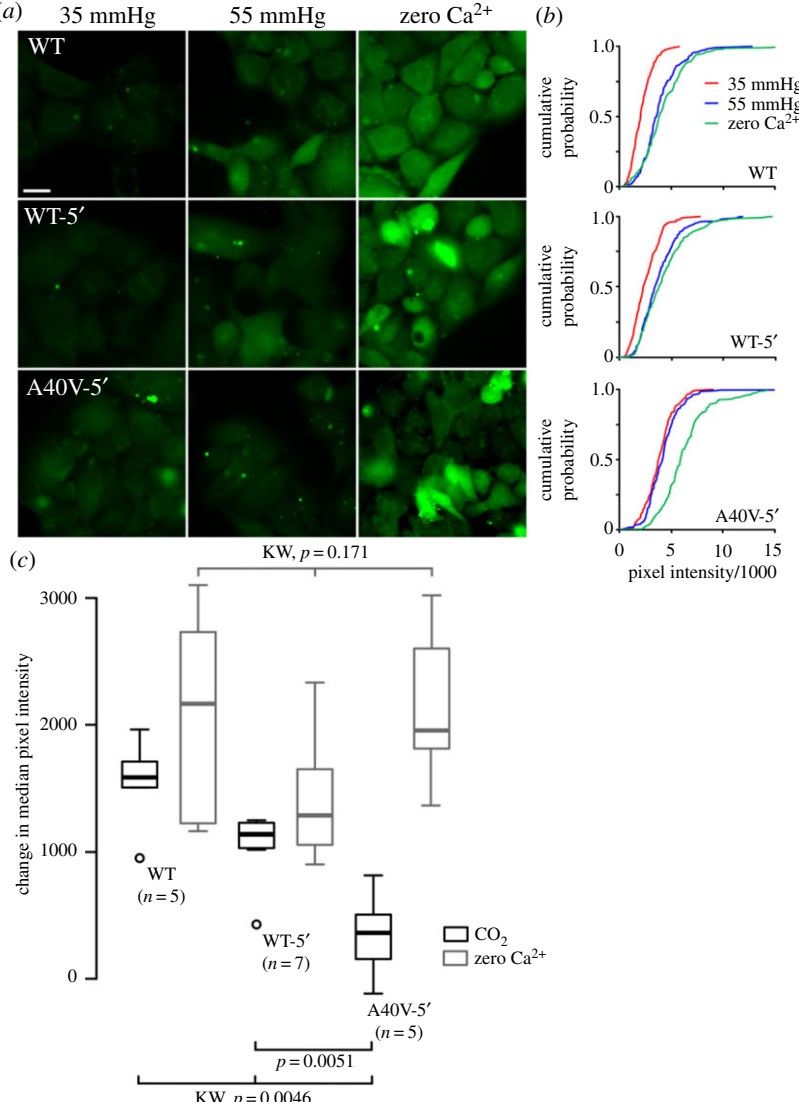

**Figure 8.** Cx26$^{A40V}$ hemichannels are insensitive to $CO_2$. As the 5′ splice mutation permits expression of this mutant, we tested the sensitivity to $CO_2$ of the Cx26$^{WT}$, Cx26$^{WT-5′}$ and Cx26$^{A40V-5′}$. (*a*) Images of dye loading at two different levels of $PCO_2$ (35 mmHg and 55 mmHg) and to a zero $Ca^{2+}$ stimulus (positive control). (*b*) Cumulative probability plots of pixel intensity show that the Cx26$^{WT-5′}$ retains sensitivity to $CO_2$, albeit slightly reduced, however, the A40V-5′ mutant channel has no sensitivity to $CO_2$. The cumulative probability distributions comprise measurements from at least five independent replicates (40 measurements per replicate). (*c*) Box and whisker plot of change in median pixel intensity from 35 to 55 mmHg of $PCO_2$ (black) and to zero $Ca^{2+}$ buffer (which opens hemichannels, grey). KW—Kruskal–Wallis ANOVA, pairwise comparison Mann–Whitney test. There is no difference in the dye loading to the zero $Ca^{2+}$ stimulus, however, the dye loading in Cx26$^{A40V-5′}$ is much less than Cx26$^{WT-5′}$. Box represents interquartile range (IQR), the central bar the median, the whiskers 1.5 times the IQR, and individual outliers are shown as circles.

## 4.2. The nature of the toxic moiety and the trans dominance of the alternative splicing

All three mutations cause the production of the same alternative product Cx26Δ. Cx26Δ lacks the 4th transmembrane domain, the second extracellular loop (required for gap junction formation) and part of the 3rd transmembrane domain. The topology of the truncated molecule is different to the full length: if it were to be expressed, the C-terminal would be extracellular instead of intracellular for the full-length molecule. We think it likely that the truncated molecule, if expressed, produces a non-functional product.

Our mutational analysis revealed that the predicted splice sites and branch point were indeed required for the production of the alternative transcripts in the mutant Cx26. Interestingly, removal of the 3′ splice site prevented the alternative splicing but did not allow good expression of the mutant Cx26. The removal of the branch points gave better expression, but the removal of the 5′ splice site

was the most effective at allowing expression of the mutant Cx26. Expression of Cx26[A88V] had a toxic effect on the HeLa cells and resulted in increased cell death (figure 4). This could be avoided by combining the A88V mutation with further mutations to remove the 3′ or 5′ splice sites or the potential branch points suggesting that some aspect of the splicing process itself, rather than the full-length A88V mutant protein, causes toxicity. Because expression of the truncated product Cx26Δ had no toxic effect, we postulate that it is the looped mRNA structure that is deleterious to the cells.

As the A88S mutation does not trigger the alternate splicing to the same degree as the KID syndrome mutations or cause cell death, the occurrence of the postulated toxic looped mRNA structure of the mutant Cx26 must be very sensitive to the precise base sequence of the mRNA. Whether analogous toxic looped mRNA structures might occur for other connexins remains an open question that would be worth investigating where expression of connexin mutants has an effect on cell survival, if only to eliminate it as a possible explanation.

Strangely, the Cx26 KID mutant mRNAs could induce alternative splicing in WT Cx26 i.e. they had a dominant effect in trans. Given that the KID syndrome mutations are always dominant, this dominant *trans* action could be pathologically significant. Whereas the mCherry or Clover C-terminal tag was necessary for the alternative splicing to occur within the transcript itself (figures 5 and 6c), a non-tagged KID syndrome mutant Cx26 could cause tagged WT-Cx26 to alternatively splice (figure 6c).

## 4.3. Keratitis ichthyosis deafness syndrome mutations and $CO_2$ sensing

We exploited our strategy of removing the 5′ splice site in Cx26[A40V] to permit good expression of this mutant to enable testing of its properties. This allowed us to show that the A40V mutation does indeed remove $CO_2$ sensitivity from the Cx26 hemichannel (figure 8). Our data now indicate that 4 out of 9 KID syndrome mutations—A88V [30], N14K, N14Y [29] and now A40V—remove $CO_2$ sensitivity from KID syndrome mutant Cx26 hemichannels. This seems beyond coincidental to us, especially as the A88S mutant has relatively mild effects on $CO_2$ sensitivity [29] and is not syndromic [34,35]. We have not yet tested the other 5 KID syndrome mutations, but it remains possible that they too have a similar effect on the $CO_2$ sensitivity of the Cx26 hemichannel. Although there is evidence that aberrant gating of mutant Cx26 hemichannels may contribute to the pathology of KID syndrome [19,21,22,24,38–40], our results strengthen the hypothesis that the actions of KID syndrome mutations on the $CO_2$ sensitivity of Cx26 may additionally contribute to its pathological syndromic effects *in vivo*.

Data accessibility. All raw quantitative data are shown in the paper (figures 4 and 8).

Authors' contributions. J.C. and N.D. designed the study, J.C. and E.d.W. performed the experiments, all authors contributed to manuscript preparation.

Competing interests. We declare we have no competing interests.

Funding. We thank the Leverhulme Trust (RPG-2015-090) and MRC (MR/P010393/1) for support.

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
