## [Reviewer comments · Royal Society Open Science]

Review History

RSOS-190188.R0 (Original submission)

Review form: Reviewer 1

Is the manuscript scientifically sound in its present form?

Yes

Are the interpretations and conclusions justified by the results?

No

Is the language acceptable?

Yes

Is it clear how to access all supporting data?

Not Applicable

Do you have any ethical concerns with this paper?

No

Have you any concerns about statistical analyses in this paper?

No

Recommendation?

Reject

Comments to the Author(s)

Brief overview of manuscript

In the present manuscript the authors have studied several mutations in GJB2 encoding the gap junction protein connexin 26 that are associated with the inflammatory skin dysplasia Keratitis Ichthyosis Deafness syndrome – KID.

They suggest that the presence of the mutation evokes an internal splice site within the mRNA that is not normally active and that this is linked with cell death. They further mutate 5' and 3' splice regions and identify a site that rescues the cytotoxicity and aberrant splicing. They also show that one of the mutations alters hemichannel sensitivity in response to CO₂. The authors conclude that the aberrant splicing is due to the addition of mCherry as a marker, however they do not in the discussion consider wider implications.

The work is well presented and aspects of the results clearly explained. However, I have several major concerns about the scope of the manuscript, the interpretation of the data in line with accumulated evidence in the field and recent literature on induction of alternative splicing linked to cell stress responses. These concerns should be reviewed and indeed further experiments warranted to relate the studies to the pathological mechanisms underlying KID syndrome.

Specific comments.

Introduction

The introduction is lacking some important information on the differences between the mutations selected for study in terms of severity of disease. Cx26A88V is established to be associated with a lethal form of KID syndrome along with the mutation Cx26G45E. Other mutations such as N14K have been reported previously and are not discussed. A similar mutation in Cx30 A88V has also recently been reported by Kelly et al (Journal of Cell Science 2019) where the lethality of this mutation is also clearly presented. Thus some further introduction to current understanding of the pathological nature of the KID mutations and the A88V locus should be included.

Methods

All the constructs used are fused in frame to the expression vector mCherry and others are in untagged vectors. HeLa Ohio cells, that do not endogenously express connexins were used in all the studies, thus the system is essentially assessing homomeric Connexin channel formation.

Transfections – cells are transiently transfected with the constructs and expression analysis assessed 48 hours -72 hours post transfection at which time there is considerable over expression of the introduced plasmids and build up of protein in intracellular stores.

Cell Death was monitored only by trypan blue exclusion assays, while more physiological assays such as a MTT assay or WST-1 assay or apoptosis assay would seem more appropriate.

Statistical analysis is appropriate

Results/Discussion

1. The time point at which the assays are performed following transient transfection is quite late. In the controls (WT and the A88S nonsyndromic mutation) there is a clear accumulation of protein in intracellular stores by 48hrs. Was a time course on mRNA extraction performed. Although A88V is not highly expressed the spatial localisation of N14K and A40V in these cell lines is not reported. Indeed previous studies have clearly shown that N14K is trapped intracellularly in the ER 24-48 hours post transfection in HeLa cells. Further assays clarifying if the cell death phenotype/ ER accumulation is associated with the alternate splicing are required.

2. Figure 2 indicates the identified internal splice sites that are activated. However the selection of the M151L mutation that rectifies this should be indicated on this figure for clarity. The

discussion should consider the implications of this on the overall folding and structure of the protein.

3. The authors clearly identify that the introduction of the M151L mutation thereby removing the 5' splice site can rectify splicing. However, there is little comment on the spatial localisation of the proteins. Indeed it looks as if the A88V protein still has trafficking issues and higher quality images with co-staining with cellular markers (e.g. ER markers) would help clarify this. This could have important implications that are not discussed.

4. The cell death assays clearly show that the removal of the splice sites restore cell viability of A88V. This is not the first mutation a A88V in a beta connexin that has been reported to cause cell death in HeLa cells. A similar mutation in Cx30, associated with Clostron syndrome was extensively studied in both HeLa cells, a cochlear cell line and a mouse model. In these studies, and others similar effects of cell death were observed in both GFP tagged and un tagged proteins. Further in the Kelly study mRNA levels of Cx30 in the mutant mice was significantly reduced where they suggest that the mutation results in loss of stability of the mRNA. Indeed there are extensive studies by this group and others that are not given recognition in this manuscript for contributions to the molecular mechanisms underlying a range of Connexin channelopathies.

5. The CO2 and hemichannel assay for A40V is tagged onto the end when most of the work has focused on the A88V mutations. It would seem more relevant to focus on A88V at this point

6. Furthermore, there is increasing literature to suggest that a number of mutations in b-connexins result in altered association with connexin43, with which the beta connexins do not normally oligomerise. It would therefore be worthy to take the A88V mutation and study the splicing effect in the presence of other connexins, providing a more integrated approach to the complex nature of connexin tissue expression profiles.

7. Finally, there is accumulating evidence that many of the connexin mutations induce ER stress pathways or mitotic crisis points, recent reports in the literature also suggest that such events can evoke alternative splicing pathways (eg Carew et al IJMS 2019). Therefore further studies should systematically rule out if the alternative splicing is truly caused by the addition of a tag or if it is a true reflection of the disease pathway.

Conclusion

While the manuscript has identified a novel pathway evoked in Hela cells expressing several KID mutations the study is premature and requires consolidation to consider the merits of the pathway in context with current literature.

Review form: Reviewer 2

Is the manuscript scientifically sound in its present form?

Yes

Are the interpretations and conclusions justified by the results?

Yes

Is the language acceptable?

Yes

Is it clear how to access all supporting data?

Yes

Do you have any ethical concerns with this paper?

No

Have you any concerns about statistical analyses in this paper?

No

Recommendation?

Accept with minor revision (please list in comments)

Comments to the Author(s)

The authors present here an interesting article that details a previously unknown phenomenon of cryptic splice site activation in disease-causing Cx26 mutants *in vitro*, which can also act in trans to cause the splicing of co-expressed wild-type Cx26. However, this feature is only evident in cells expressing tagged Cx26 and therefore does not occur *in vivo*, or in cells expressing untagged Cx26/mutants. The likely pathological mechanism in KID syndrome is related to the reduction of hemichannel CO₂ sensitivity which the authors have found across several KID mutants – including now the A40V mutant in this study. The manuscript is well written, and the authors have revealed a detrimental mechanism of tagged Cx26 mutants *in vitro*, which should be taken into consideration for future research in the connexin field.

I only have the following minor corrections and considerations for the authors to address:

- 1) Insert the word “is” before “unavoidable” online 29 of abstract
- 2) What’s the scale of the Y axis in Fig.4, - is it percentage? If so, it should be written as such in the axis label.
- 3) Splice is spelled as “spice” on page 11, line 7.
- 4) Is the untagged Cx26A88V mutant toxic to cells alone? It would also be interesting to see additional data of untagged mutant in Figure 4.
- 5) It’s interesting that untagged input and reporter Cx26/mutants do not get spliced, whereas tagged versions so. Therefore, I think it would be useful to see the untagged Cx26A88V and untagged Cx26WT data in Fig. 6 rather than saying “(data not shown)”.
- 6) Although untagged Cx26 mutants lack the ability to undergo aberrant splicing (i.e. it only occurs with tagged proteins *in vitro*), is there any evidence that the mutations could have an affect on other connexins? We know that Cx30 is often expressed in the same cell types as Cx26 and it’s interesting that some mutations in Cx26 are mirrored in Cx30 in terms of their location/substitution (such as A88V) and phenotype (skin disease and/or hearing loss). A recent article by Kelly et al. (J. Cell Sci., 2019) found that the untagged Cx30 A88V mutant killed cells *in vitro*, but not *in vivo*, and reduced both mRNA and protein expression of Cx30 in the cochleae of Cx30A88V mutant mice. Have the authors have looked at this mechanism with other connexin subtypes? Or could the authors comment on the possible trans effects across connexin subtypes?

Decision letter (RSOS-190188.R0)

29-Apr-2019

Dear Professor Dale:

Manuscript ID RSOS-190188 entitled "Cx26 KID syndrome mutations trigger alternative splicing

of Cx26 to prevent expression and cause toxicity in vitro" which you submitted to Royal Society Open Science, has been reviewed. The comments from reviewers are included at the bottom of this letter.

In view of the criticisms of the reviewers, the manuscript has been rejected in its current form. However, a new manuscript may be submitted which takes into consideration these comments - the Editors have opted to give you a reject/resubmit decision to allow you the fullest opportunity to tackle their concerns.

Please note that resubmitting your manuscript does not guarantee eventual acceptance, and that your resubmission will be subject to peer review before a decision is made.

Your resubmitted manuscript should be submitted by 27-Oct-2019. If you are unable to submit by this date please contact the Editorial Office.

on behalf of Dr John Dalton (Associate Editor) and Catrin Pritchard (Subject Editor)
openscience@royalsociety.org

Editor comments:

The reviewers and AE agree that this is a potentially interesting manuscript but have raised a number of major flaws which need to be addressed. These could potentially be addressed in a re-submission but the manuscript as it stands cannot be accepted.

Associate Editor Comments to Author (Dr John Dalton):

Your manuscript was deemed original and interesting by our reviewers. However, one reviewer raised substantial concerns and suggested further interpretation, attention to other studies in the field and, indeed, some further experiments to relate the studies to the pathological mechanisms underlying KID syndrome.

Reviewers' Comments to Author:

Reviewer: 1

Comments to the Author(s)

Brief overview of manuscript

In the present manuscript the authors have studied several mutations in GJB2 encoding the gap junction protein connexin 26 that are associated with the inflammatory skin dysplasia Keratitis Ichthyosis Deafness syndrome – KID.

They suggest that the presence of the mutation evokes an internal splice site within the mRNA that is not normally active and that this is linked with cell death. They further mutate 5' and 3' splice regions and identify a site that rescues the cytotoxicity and aberrant splicing. They also show that one of the mutations alters hemichannel sensitivity in response to CO₂. The authors conclude that the aberrant splicing is due to the addition of mcherry as a marker, however they do not in the discussion consider wider implications.

The work is well presented and aspects of the results clearly explained. However, I have several major concerns about the scope of the manuscript, the interpretation of the data in line with accumulated evidence in the field and recent literature on induction of alternative splicing linked to cell stress responses. These concerns should be reviewed and indeed further experiments warranted to relate the studies to the pathological mechanisms underlying KID syndrome.

Specific comments.

Introduction

The introduction is lacking some important information on the differences between the mutations selected for study in terms of severity of disease. Cx26A88V is established to be associated with a lethal form of KID syndrome along with the mutation Cx26G45E. Other mutations such as N14K have been reported previously and are not discussed. A similar mutation in Cx30 A88V has also recently been reported by Kelly et al (Journal of Cell Science 2019) where the lethality of this mutation is also clearly presented. Thus some further introduction to current understanding of the pathological nature of the KID mutations and the A88V locus should be included.

Methods

All the constructs used are fused in frame to the expression vector mCherry and others are in untagged vectors. HeLa Ohio cells, that do not endogenously express connexins were used in all the studies, thus the system is essentially assessing homomeric Connexin channel formation. Transfections – cells are transiently transfected with the constructs and expression analysis assessed 48 hours -72 hours post transfection at which time there is considerable over expression of the introduced plasmids and build up of protein in intracellular stores.

Cell Death was monitored only by trypan blue exclusion assays, while more physiological assays such as a MTT assay or WST-1 assay or apoptosis assay would seem more appropriate.

Statistical analysis is appropriate

Results/Discussion

1. The time point at which the assays are performed following transient transfection is quite late. In the controls (WT and the A88S nonsyndromic mutation) there is a clear accumulation of protein in intracellular stores by 48hrs. Was a time course on mRNA extraction performed. Although A88V is not highly expressed the spatial localisation of N14K and A40V in these cell lines is not reported. Indeed previous studies have clearly shown that N14K is trapped intracellularly in the ER 24-48 hours post transfection in HeLa cells. Further assays clarifying if the cell death phenotype/ ER accumulation is associated with the alternate splicing are required.

2. Figure 2 indicates the identified internal splice sites that are activated. However the selection of the M151L mutation that rectifies this should be indicated on this figure for clarity. The discussion should consider the implications of this on the overall folding and structure of the protein.

3. The authors clearly identify that the introduction of the M151L mutation thereby removing the 5' splice site can rectify splicing. However, there is little comment on the spatial localisation of the proteins. Indeed it looks as if the A88V protein still has trafficking issues and higher quality images with co-staining with cellular markers (e.g. ER markers) would help clarify this. This could have important implications that are not discussed.

4. The cell death assays clearly show that the removal of the splice sites restore cell viability of A88V. This is not the first mutation a A88V in a beta connexin that has been reported to cause cell death in HeLa cells. A similar mutation in Cx30, associated with Clostron syndrome was extensively studied in both HeLa cells, a cochlear cell line and a mouse model. In these studies, and others similar effects of cell death were observed in both GFP tagged and un tagged proteins. Further in the Kelly study mRNA levels of Cx30 in the mutant mice was significantly reduced where they suggest that the mutation results in loss of stability of the mRNA. Indeed there are extensive studies by this group and others that are not given recognition in this manuscript for contributions to the molecular mechanisms underlying a range of Connexin channelopathies.
5. The CO2 and hemichannel assay for A40V is tagged onto the end when most of the work has focused on the A88V mutations. It would seem more relevant to focus on A88V at this point
6. Furthermore, there is increasing literature to suggest that a number of mutations in b-connexins result in altered association with connexin43, with which the beta connexins do not normally oligomerise. It would therefore be worthy to take the A88V mutation and study the splicing effect in the presence of other connexins, providing a more integrated approach to the complex nature of connexin tissue expression profiles.
7. Finally, there is accumulating evidence that many of the connexin mutations induce ER stress pathways or mitotic crisis points, recent reports in the literature also suggest that such events can evoke alternative splicing pathways (eg Carew et al IJMS 2019). Therefore further studies should systematically rule out if the alternative splicing is truly caused by the addition of a tag or if it is a true reflection of the disease pathway.

Conclusion

While the manuscript has identified a novel pathway evoked in Hela cells expressing several KID mutations the study is premature and requires consolidation to consider the merits of the pathway in context with current literature.

Reviewer: 2

Comments to the Author(s)

The authors present here an interesting article that details a previously unknown phenomenon of cryptic splice site activation in disease-causing Cx26 mutants *in vitro*, which can also act in trans to cause the splicing of co-expressed wild-type Cx26. However, this feature is only evident in cells expressing tagged Cx26 and therefore does not occur *in vivo*, or in cells expressing untagged Cx26/mutants. The likely pathological mechanism in KID syndrome is related to the reduction of hemichannel CO2 sensitivity which the authors have found across several KID mutants – including now the A40V mutant in this study. The manuscript is well written, and the authors have revealed a detrimental mechanism of tagged Cx26 mutants *in vitro*, which should be taken into consideration for future research in the connexin field.

I only have the following minor corrections and considerations for the authors to address:

- 1) Insert the word “is” before “unavoidable” online 29 of abstract
- 2) What’s the scale of the Y axis in Fig.4, - is it percentage? If so, it should be written as such in the axis label.
- 3) Splice is spelled as “spice” on page 11, line 7.
- 4) Is the untagged Cx26A88V mutant toxic to cells alone? It would also be interesting to see additional data of untagged mutant in Figure 4.
- 5) It’s interesting that untagged input and reporter Cx26/mutants do not get spliced, whereas tagged versions so. Therefore, I think it would be useful to see the untagged Cx26A88V and untagged Cx26WT data in Fig. 6 rather than saying “(data not shown)”.

6) Although untagged Cx26 mutants lack the ability to undergo aberrant splicing (i.e. it only occurs with tagged proteins in vitro), is there any evidence that the mutations could have an affect on other connexins? We know that Cx30 is often expressed in the same cell types as Cx26 and it's interesting that some mutations in Cx26 are mirrored in Cx30 in terms of their location/substitution (such as A88V) and phenotype (skin disease and/or hearing loss). A recent article by Kelly et al. (J. Cell Sci., 2019) found that the untagged Cx30 A88V mutant killed cells in vitro, but not in vivo, and reduced both mRNA and protein expression of Cx30 in the cochleae of Cx30A88V mutant mice. Have the authors have looked at this mechanism with other connexin subtypes? Or could the authors comment on the possible trans effects across connexin subtypes?

Author's Response to Decision Letter for (RSOS-190188.R0)

See Appendix A.

Decision letter (RSOS-191128.R0)

08-Jul-2019

Dear Professor Dale,

I am pleased to inform you that your manuscript entitled "Cx26 KID syndrome mutations trigger alternative splicing of Cx26 to prevent expression and cause toxicity in vitro" is now accepted for publication in Royal Society Open Science.

You have the opportunity to archive your accepted, unbranded manuscript, but access to the full text must be embargoed until publication.

Articles are normally press released. For this to be effective we set an embargo on news coverage corresponding to the publication date of the article. We request that news media and the authors do not publish stories ahead of this embargo (when final version of the article is available).

Kind regards,
Alice Power
Editorial Coordinator
Royal Society Open Science

on behalf of Dr John Dalton (Associate Editor) and Catrin Pritchard (Subject Editor)
openscience@royalsociety.org

Appendix A

Editor comments:

The reviewers and AE agree that this is a potentially interesting manuscript but have raised a number of major flaws which need to be addressed. These could potentially be addressed in a re-submission but the manuscript as it stands cannot be accepted.

Associate Editor Comments to Author (Dr John Dalton):

Your manuscript was deemed original and interesting by our reviewers. However, one reviewer raised substantial concerns and suggested further interpretation, attention to other studies in the field and, indeed, some further experiments to relate the studies to the pathological mechanisms underlying KID syndrome.

We agree that adding further reference to other studies in the field, and improving the Discussion strengthens the paper. Much of the comments from Reviewer 1 point to a desire to answer questions that this paper was not designed to investigate and are not relevant to the novel phenomenon that we have discovered. We think it worth being clear up front that while we have responded as much as we can to the Reviewers' comments and have added some new data, we are not willing to perform wholesale new experiments on trafficking or pathology in response to Reviewer 1 for the reasons given below.

Reviewer: 1

Comments to the Author(s)

Brief overview of manuscript

In the present manuscript the authors have studied several mutations in GJB2 encoding the gap junction protein connexin 26 that are associated with the inflammatory skin dysplasia Keratitis Ichthyosis Deafness syndrome KID. They suggest that the presence of the mutation evokes an internal splice site within the mRNA that is not normally active and that this is linked with cell death. They further mutate 5' and 3' splice regions and identify a site that rescues the cytotoxicity and aberrant splicing. They also show that one of the mutations alters hemichannel sensitivity in response to CO₂. The authors conclude that the aberrant splicing is due to the addition of mcherry as a marker, however they do not in the discussion consider wider implications.

The work is well presented and aspects of the results clearly explained. However, I have several major concerns about the scope of the manuscript, the interpretation of the data in line with accumulated evidence in the field and recent literature on induction of alternative splicing linked to cell stress responses. These concerns should be reviewed and indeed further experiments warranted to relate the studies to the pathological mechanisms underlying KID syndrome.

Specific comments.

Introduction

The introduction is lacking some important information on the differences between the mutations selected for study in terms of severity of disease. Cx26A88V is established to be associated with a lethal form of KID syndrome along with the mutation Cx26G45E. Other mutations such as N14K have been reported previously and are not discussed. A similar mutation in Cx30 A88V has also recently been reported by Kelly et al (Journal of Cell Science 2019) where the lethality of this mutation is also clearly presented. Thus some further introduction to current understanding of the pathological nature of the KID mutations and the A88V locus should be included.

We have added greater explanation of KID syndrome mutations and references to the introduction pp 3.

Methods

All the constructs used are fused in frame to the expression vector mCherry and others are in untagged vectors. HeLa Ohio cells, that do not endogenously express connexins were used in all the studies, thus the system is essentially assessing homomeric Connexin channel formation.

Transfections – cells are transiently transfected with the constructs and expression analysis assessed 48 hours - 72 hours post transfection at which time there is considerable over expression of the introduced plasmids and build up of protein in intracellular stores.

Cell Death was monitored only by trypan blue exclusion assays, while more physiological assays such as a MTT assay or WST-1 assay or apoptosis assay would seem more appropriate.

In our view the Trypan Blue assay quantifies cell death and is informative -it shows that it is the splicing itself, but not the spliced product, that is the important factor in increasing cell death. We do not see that we need to use another assay to document this point.

Statistical analysis is appropriate

Results/Discussion

1. The time point at which the assays are performed following transient transfection is quite late. In the controls (WT and the A88S nonsyndromic mutation) there is a clear accumulation of protein in intracellular stores by 48hrs. Was a time course on mRNA extraction performed. Although A88V is not highly expressed the spatial localisation of N14K and A40V in these cell lines is not reported. Indeed previous studies have clearly shown that N14K is trapped intracellularly in the ER 24-48 hours post transfection in HeLa cells. Further assays clarifying if the cell death phenotype/ ER accumulation is associated with the alternate splicing are required.

mRNA analysis was performed at 48h post transfection. Our aim was to test for the presence of the mRNA because expression levels of the mutant connexins were so poor. mRNA extractions at different time points were not seen as informative in this regard. We performed the assays at a time point when we know that functional Cx26 hemichannels (WT and mutant) are present in the plasma membrane from our previous studies (including N14K/N14Y, de Wolf et al 2016, *Physiol Rep* 4, e13038). Given that the toxic moiety seems to be the folded mRNA and not the protein and the alternatively spliced product is non-functional, accumulation in the ER seems a moot point to us.

2. Figure 2 indicates the identified internal splice sites that are activated. However the selection of the M151L mutation that rectifies this should be indicated on this figure for clarity. The discussion should consider the implications of this on the overall folding and structure of the protein.

We have added a notation to the legend of Figure 2 that the highlighted codon at the 5' splice site is for Met151. We have added material to the discussion (pp 11). M151 is an outward facing residue of the third transmembrane domain that interacts with the membrane lipids, making it unlikely that there is a substantial change in channel conformation. WT Cx26 with the conservative M151L mutation is still CO₂ sensitive, still permeable to the fluorescent dye and can still be blocked by extracellular Ca²⁺. By these measures the channels still retains the major functions necessary for its role as a CO₂ sensitive hemichannel.

3. The authors clearly identify that the introduction of the M151L mutation thereby removing the 5' splice site can rectify splicing. However, there is little comment on the spatial localisation of the proteins. Indeed it looks as if the A88V protein still has trafficking issues and higher quality images with co-staining with cellular markers (e.g. ER markers) would help clarify this. This could have important implications that are not discussed.

This paper is about the splicing not about the trafficking. The M151L mutation rescues expression minimizes cell death and allows functional hemichannels to enter the plasma membrane. This is true also for A88V which exhibits CO₂ insensitive hemichannels, an observation that we did not include in this paper as we had already documented this previously.

4. The cell death assays clearly show that the removal of the splice sites restore cell viability of A88V. This is not the first mutation a A88V in a beta connexin that has been reported to cause cell death in HeLa cells. A similar mutation in Cx30, associated with Clostron syndrome was extensively studied in both HeLa cells, a cochlear cell line and a mouse model. In these studies, and others similar effects of cell death were observed in both GFP tagged and un tagged proteins. Further in the Kelly study mRNA levels of Cx30 in the mutant mice was significantly reduced where they suggest that the mutation results in loss of stability of the mRNA. Indeed there are extensive studies by this group and others that are not given recognition in this manuscript for contributions to the molecular mechanisms underlying a range of Connexin channelopathies.

This is a different connexin with a very different C-terminal tail. The relevance of this work is not immediately obvious to our work on Cx26, but we have made reference to this with brief

discussion on pp 8. As mentioned earlier we have included a more comprehensive section in the introduction on mutations and channelopathies relevant to Cx26 (pp 3).

5. The CO₂ and hemichannel assay for A40V is tagged onto the end when most of the work has focused on the A88V mutations. It would seem more relevant to focus on A88V at this point

We have already published the CO₂ insensitivity of A88V, we chose A40V to extend knowledge of the effects of KID syndrome mutations.

6. Furthermore, there is increasing literature to suggest that a number of mutations in b-connexins result in altered association with connexin43, with which the beta connexins do not normally oligomerise. It would therefore be worthy to take the A88V mutation and study the splicing effect in the presence of other connexins, providing a more integrated approach to the complex nature of connexin tissue expression profiles.

This is beyond the scope of what we are reporting in this paper. We also point out that the alternatively spliced product has an altered membrane topology (C-terminus now extracellular) so is very likely non-functional (pp11, “The nature of the toxic moiety and the trans dominance of the alternative splicing”).

7. Finally, there is accumulating evidence that many of the connexin mutations induce ER stress pathways or mitotic crisis points, recent reports in the literature also suggest that such events can evoke alternative splicing pathways (eg Carew et al IJMS 2019). Therefore further studies should systematically rule out if the alternative splicing is truly caused by the addition of a tag or if it is a true reflection of the disease pathway.

We think there is little doubt that the alternate splicing is caused by the addition of the tag. Our additional data in Figure 6 makes this very clear (see Fig 6C and D -where there is no tag there is no splicing).

However, it is a very interesting question as to whether this could occur in the human disease pathway. It is entirely possible that the non-coding sequences of the mRNA (missing from our expression constructs) could cause the splicing to occur for the KID syndrome mutations. We discuss this in the paper. The best and most direct way to answer this question would be to extract mRNA from human KID syndrome patients. We think this is an important issue to be settled by others with access to relevant fresh tissue samples and our paper may provide sufficient motivation for them to do this.

Conclusion

While the manuscript has identified a novel pathway evoked in Hela cells expressing several KID mutations the study is premature and requires consolidation to consider the merits of the pathway in context with current literature.

We disagree that our study is premature. We have been very open about the limits of our study, and whether this is pathophysiologically relevant. We think this is an interesting cellular phenomenon that we discovered by chance. It is worth reporting as many investigators in the field still tag their mutant connexins with a fluorescent protein. Our paper will assist them to design their experimental approaches to achieve better cellular expression and avoid a potentially artefactual confounding factor.

Reviewer: 2

Comments to the Author(s)

The authors present here an interesting article that details a previously unknown phenomenon of cryptic splice site activation in disease-causing Cx26 mutants *in vitro*, which can also act in trans to cause the splicing of co-expressed wild-type Cx26. However, this feature is only evident in cells expressing tagged Cx26 and therefore does not occur *in vivo*, or in cells expressing untagged Cx26/mutants. The likely pathological mechanism in KID syndrome is related to the reduction of hemichannel CO₂ sensitivity which the authors have found across several KID mutants – including now the A40V mutant in this study. The manuscript is well written, and the authors have revealed a detrimental mechanism of tagged Cx26 mutants *in vitro*, which should be taken into consideration for future research in the connexin field.

I only have the following minor corrections and considerations for the authors to address:

1) Insert the word “is” before “unavoidable” online 29 of abstract

This has been corrected

2) What’s the scale of the Y axis in Fig.4, - is it percentage? If so, it should be written as such in the axis label.

This is now clarified in the axis label -it is fold increase over mock-transfected control.

3) Splice is spelled as “spice” on page 11, line 7.

This has been corrected

4) Is the untagged Cx26A88V mutant toxic to cells alone? It would also be interesting to see additional data of untagged mutant in Figure 4.

We have added these new data to Figure 4.

5) It’s interesting that untagged input and reporter Cx26/mutants do not get spliced, whereas tagged versions so. Therefore, I think it would be useful to see the untagged Cx26A88V and untagged Cx26WT data in Fig. 6 rather than saying “(data not shown)”.

We now show this result in a new version of Figure 6.

6) Although untagged Cx26 mutants lack the ability to undergo aberrant splicing (i.e. it only occurs with tagged proteins in vitro), is there any evidence that the mutations could have an effect on other connexins? We know that Cx30 is often expressed in the same cell types as Cx26 and it’s interesting that some mutations in Cx26 are mirrored in Cx30 in terms of their location/substitution (such as A88V) and phenotype (skin disease and/or hearing loss). A recent article by Kelly et al. (J. Cell Sci., 2019) found that the untagged Cx30 A88V mutant killed cells in vitro, but not in vivo, and reduced both mRNA and protein expression of Cx30 in the cochleae of Cx30A88V mutant mice. Have the authors have looked at this mechanism with other connexin subtypes? Or could the authors comment on the possible trans effects across connexin subtypes?

We have not looked at other connexin subtypes. Given this appears to be a mechanism that depends very specifically on the mRNA sequence (A88V but not A88S triggers this) it may be unlikely that different genes with different nucleotide sequences would display this, but of course it might still occur. We have added some discussion of this on pp 12.